# Palmitoylation Is Indispensable for Remorin to Restrict Tobacco Mosaic Virus Cell-to-Cell Movement in *Nicotiana benthamiana*

**DOI:** 10.3390/v14061324

**Published:** 2022-06-17

**Authors:** Tingting Ma, Shuai Fu, Kun Wang, Yaqin Wang, Jianxiang Wu, Xueping Zhou

**Affiliations:** 1State Key Laboratory of Rice Biology, Institute of Biotechnology, Zhejiang University, Hangzhou 310058, China; 11716069@zju.edu.cn (T.M.); fushuai@zju.edu.cn (S.F.); kunwang@zju.edu.cn (K.W.); yaqinwang@zju.edu.cn (Y.W.); wujx@zju.edu.cn (J.W.); 2State Key Laboratory for Biology of Plant Diseases and Insect Pests, Institute of Plant Protection, Chinese Academy of Agricultural Sciences, Beijing 100193, China

**Keywords:** remorin, palmitoylation, movement, tobacco mosaic virus, plasma membrane, plasmodesmata

## Abstract

Remorin (REM) is a plant-specific plasma membrane-associated protein regulating plasmodesmata plasticity and restricting viral cell-to-cell movement. Here, we show that palmitoylation is broadly present in group 1 remorin proteins in *Nicotiana benthamiana* and is crucial for plasma membrane localization and accumulation. By screening the four members of *N. benthamiana* group 1 remorin proteins, we found that only NbREM1.5 could significantly hamper tobacco mosaic virus (TMV) cell-to-cell movement. We further showed that NbREM1.5 interacts with the movement protein of TMV in vivo and interferes with its function of expanding the plasmodesmata size exclusion limit. We also demonstrated that palmitoylation is indispensable for NbREM1.5 to hamper plasmodesmata permeability and inhibit TMV cell-to-cell movement.

## 1. Introduction

Plasmodesmata (PD) are plant-specific plastic membranous nanopores connecting adjacent plant cells [1,2]. PD are communication channels for the intercellular transport of proteins and molecules, and they are utilized by plant viruses to establish intercellular trafficking [3,4]. The efficiency of cell-to-cell communication and transportation through PD is dependent on the PD size exclusion limit (SEL), which is strictly controlled by callose deposition in the PD neck region [5]. As obligate intracellular pathogens, plant viruses encode movement proteins (MPs) to expand PD permeability and promote viral cell-to-cell movement [6]. Several plant proteins are also found capable of controlling PD SEL, of which, the remorin family is one of the best-characterized plant plasma membrane-anchored proteins regulating plasmodesmata aperture and functionality [7,8,9]. As nanodomain-organized proteins located at the plasma membrane and PD, remorin proteins are found to be broadly involved in plant responses to biotic stress, including viruses, bacteria, fungi, oomycetes, etc. [10].

Although remorin proteins are specifically associated with the inner leaflet of the plasma membrane, they lack a transmembrane domain. The molecular mechanism for remorin proteins to specifically anchor at the membrane remains unclear [10]. The conserved remorin C-terminal anchor region (REM-CA) of potato (*Solanum tuberosum*) remorin group 1 isoform 3 (StREM1.3) is considered to be a strictly membrane-anchoring motif, by folding into an alpha helix and inserting itself into the hydrophobic core of the bilayer [11,12,13]. For most *Arabidopsis* remorin proteins, palmitoylation of cysteine residues in the C-terminal region has been shown to predominantly contribute to their membrane association but it is not a key determinant factor for their membrane nanodomain localization [14].

Palmitoylation, inter-changeably used with the term “*S*-acylation”, is a type of lipid post-translational modification. Palmitoylation is the reversible addition of a long-chain fatty acid (16-carbon palmitate) to a specific cysteine residue via a thioester bond [15,16,17]. *Oryza sativa* GSD1 (rice remorin group 6 isoform) attaches to the plasma membrane through palmitoylation to regulate plasmodesmata conductance for photoassimilate transport [18]. Our previous study revealed that rice stripe virus (RSV) interferes with the palmitoylation of *Oryza sativa* and *Nicotiana benthamiana* remorin group 1 isoforms (OsREM1.4 and NbREM1) resulting in remorin proteins degraded through autophagy, and finally facilitating RSV cell-to-cell movement [8].

Although palmitoylation is well characterized, contributing to remorin protein localization to the plasma membrane and the performance of various functions, the reports on the palmitoylation of remorin regulating plant virus infections are very limited. Here, by using *N. benthamiana* group 1 remorin isoforms and one of the best-characterized positive-sense single-stranded RNA viruses, tobacco mosaic virus (TMV), we verify that NbREM1.5 is the only member of the group 1 remorin proteins in *N. benthamiana* that can significantly suppress the intercellular movement of TMV, and palmitoylation is indispensable for NbREM1.5 to restrict TMV cell-to-cell movement.

## 2. Materials and Methods

### 2.1. Plant Growth Conditions

The *N. benthamiana* plants were grown in the growth chamber at 25 °C under a 16-h light/8-h dark photoperiod with 60% relative humidity.

### 2.2. RNA Extraction and Plasmid Construction

Total RNA was extracted from five/seven-leaf stage *N. benthamiana* plant leaves by using TRIzol reagent (Invitrogen, Carlsbad, CA, USA) according to the manufacturer’s instructions. *N. benthamiana* complementary DNA (cDNA) was synthesized by reverse transcription of *N. benthamiana* leaf total RNA using ReverTra Ace qPCR RT Master Mix with gDNA Remover (TOYOBO, Osaka, Japan) according to the manufacturer’s instructions. The specific primer pairs GFP/Flag-NbREM1.1-Xba1-F/R, GFP/Flag-NbREM1.3-Xba1-F/R, GFP/Flag-NbREM1.5-Xba1-F/R and GFP/Flag-NbREM1.8-Xba1-F/R were used, respectively, to amplify the full-length ORFs of *NbREM1.1* and *NbREM1.3*, *NbREM1.5* and *NbREM1.8* from *N. benthamiana* cDNA. The sequences of the amplified *NbREM1.1*, *NbREM1.3*, *NbREM1.5* and *NbREM1.8* were deposited in GenBank under accession numbers ON392760 (*NbREM1.1*), ON392761 (*NbREM1.3*), ON392762 (*NbREM1. 5*) and ON392763 (*NbREM1.8*). Site-specific mutation of cysteine to alanine in *NbREM1.1*, *NbREM1.3*, *NbREM1.5* or *NbREM1.8* was generated by introducing a mutation into a reverse primer pair GFP/Flag-NbREM1.1-C206A-Xba1-R, GFP/Flag-NbREM1.3-C177A-Xba1-R, GFP/Flag-NbREM1.5-C172A-Xba1-R, GFP/Flag-NbREM1.5-C172/175A-Xba1-R or GFP/Flag-NbREM1.8-C195A-Xba1-R), then the primer pairs were used for amplification. For the construction of N-terminus GFP or flag-tagged NbREM1.1, NbREM1.3, NbREM1.5 and NbREM1.8, and their palmitoylation-deficient mutants, pCAMBIA1300-2×35S::GFP::MCS or pCAMBIA1300-2×35S::Flag::MCS vector was digested by *Xba1*, then *NbREM1.1, NbREM1.1-C206A, NbREM1.3, NbREM1.3-C177A, NbREM1.5, NbREM1.5-C172A, NbREM1.5-C172/175A, NbREM1.8* or *NbREM1.8-C195A* was fused, respectively, into pCAMBIA1300-2×35S::GFP::MCS or pCAMBIA1300-2×35S::Flag::MCS by using the ClonExpress II One Step Cloning Kit (Vazyme, Nanjing, China) according to the manufacturer’s instructions. The monomeric red fluorescent protein mScarlet was used as a negative control protein for agroinfiltration [19]. The mScarlet sequence was codon optimized and synthesized for expression in *Arabidopsis thaliana* and *N**. benthamiana* (GenScript, Nanjing, China). The β-glucuronidase (GUS) was amplified from the pENTR-GUS plasmid using GUS-Flag-Sma1-F/R primer pairs. For the construction of mScarlet-flag or GUS-flag plasmid, pCAMBIA1300-2×35S::Flag::MCS was digested by *Sma1*, then *mScarlet* or GUS-flag was fused into pCAMBIA1300-2×35S::Flag::MCS by using the ClonExpress II One Step Cloning Kit (Vazyme, Nanjing, China). Primers used for cloning in this study are listed in Appendix A.

### 2.3. Agroinfiltration in N. benthamiana

For agroinfiltration, vectors were individually transformed into *A. tumefaciens* strain EHA105 cells, and then the transformed *A**. tumefaciens* cells were grown individually in YEP medium containing 50 mg/L kanamycin and 50 mg/L rifampicin at 28 °C until an OD_600_ of 0.6 was reached. The cultures were collected and re-suspended in the inoculation buffer (10 mM MgCl_2_, 100 mM MES (pH 5.7), 2 mM acetosyringone) at room temperature. The suspensions were adjusted to an OD_600_ of 0.5 and then used for agroinfiltration in five/seven-leaf *N. benthamiana* plants.

### 2.4. Multiple Sequence Alignment and Phylogenetic Analysis

Multiple sequence alignment was analyzed through the Clustal W method (Appendix A) [20]. Phylogenetic analysis was performed by MEGA X [21]. The evolutionary history was inferred by using the maximum likelihood method and an JTT matrix-based model [21]. The tree was tested via bootstrap analysis (1000 replicates) and was drawn to scale (Appendix A).

### 2.5. Reverse Transcription-Quantitative PCR (RT-qPCR)

Primer pairs RT-NbREM1.3-F/R, RT-NbREM1.7-F/R, and RT-NbREM1.9-F/R specific to *NbREM1.3*, *NbREM1.7* and *NbREM1.9* transcripts, respectively, were designed by using SnapGene (Insightful Science, snapgene.com, accessed on 1 March 2020). Since they contained a high sequence identity, the clade specific primer pairs RT-NbREM1.1/1.2-F/R, RT-NbREM1.4/1.5-F/R and RT-NbREM1.6/1.8-F/R, were designed to amplify *NbREM1.1/NbREM1.2*, *NbREM1.4*/*NbREM1.5* and *NbREM1.6*/*NbREM1.8*, respectively. *N. benthamiana actin* (*NbActin*, Niben101Scf03410g03002.1) was used as a reference gene for determining the relative expression of *NbREM1.1*/*NbREM1.2*, *NbREM1.4*/*NbREM1.5*, *NbREM1.6*/*NbREM1.8*, *NbREM1.3*, *NbREM1.7* and *NbREM1.9* transcripts. Primer pair RT-NbActin-F/R was designed to amplify the *NbActin*. RT-qPCR was performed using a ChamQ SYBR qPCR master Mix (Vazyme, Nanjing, China) in LightCycler 480 (Roche, Rotkreuz, Switzerland). The RT-qPCR conditions consisted of 95 °C for 30 s followed by 40 cycles of 95 °C for 10 s, 60 °C for 30 s. The intensity of SYBR Green I fluorescence was measured at the end of each cycle. The cycle threshold (Ct), in which the fluorescent signal reached a threshold value was determined using software provided with the LightCycler 480. The amplification specificity was validated using a melting curve analysis, which consisted of denaturation at 95 °C for 15 s and annealing at 60 °C at 60 s, followed by continuous measurement of fluorescence intensity at increasing temperatures of 0.1 °C per second until the temperature reached 95 °C. The relative quantities of NbREMs transcripts were calculated according to the ΔΔCt method in LightCycler 480 (Roche, Rotkreuz, Switzerland). Primers are listed in Appendix A.

### 2.6. Virus Inoculation and Detection

The TMV-GFP infectious clone (30B-GFPC3) [22] was infiltrated into *N. benthamiana* leaves by *A. tumefaciens* infiltration; infection fluorescence spots were observed and photographed under a portable UV lamp at 4 days post-infiltration (dpi) on the inoculated leaves. TMV-GFP infection spot areas were calculated by Image Pro Plus 6.0 software (Media Cybernetics, Silver Spring, MD, USA) and were analyzed by GraphPad Prism version 8.0 (GraphPad Software, San Diego, CA, USA). Experiments were repeated at least three times. Statistical parameters SEM (standard error) are indicated in figure legends. Asterisks mark statistical significance (* 0.01 < *p* < 0.05; ** *p* < 0.01; *** *p* < 0.001; ns, *p* > 0.05, not significant) according to two-tailed Student’s *t*-test.

### 2.7. Acyl-Resin Assisted Capture (Acyl-RAC)

The experiment was performed according to the method previously reported [23]. Briefly, *N. benthamiana* leaves that have expressed proteins of interest by agroinfiltration were harvested at 48 h post infiltration (hpi). Total proteins were extracted with a lysis buffer (25 mM HEPES, 25 mM NaCl, 1 mM EDTA, PH 7.5) containing a protease inhibitor cocktail (Roche, Rotkreuz, Switzerland). Total protein extraction was diluted with one volume of blocking buffer (100 mM HEPES, 2.5% SDS, 1 mM EDTA, 0.5% MMTS, PH 7.5), followed by incubating at 40 °C for 30 min to block free thiols. Then, three volumes of cold acetone were added, and samples were allowed to be precipitated at −20 °C for 20 min. The precipitated proteins were collected by centrifugation at 5000× *g* for 10 min. The pellet was then washed with 70% cold acetone, and finally re-suspended with a binding buffer (100 mM HEPES, 1% SDS, 1 mM EDTA, PH 7.5). The samples were split into two equal parts followed by adding 60 µL of pre-washed thiopropyl Sepharose 6b resins (Sigma, St. Louis, MO, USA). Then, 40 µL of 2 M neutral hydroxylamine (NH_2_OH) (pH 7.5) or 2 M NaCl (used as a negative control) were added, respectively, into two equal parts of the mix and processed in parallel. A total of 100 μL of each mix was saved as the total input (input). The mixtures were rotated at room temperature for 2 h then centrifuged at 5000× *g* for 1 min; the pellets containing thiopropyl Sepharose 6b resins were washed 5 times using the binding buffer, and finally, the resins were suspended in 100 µL binding buffer containing 50 mM dithiothreitol (DTT). A total of 2 × SDS-PAGE loading buffer was added to the samples and the samples were heated to 100 °C for 10 min, followed by immunoblotting detection.

### 2.8. Immunoblotting and Antibodies

The immunoblotting was performed essentially as described previously [24]. Briefly, plant total proteins were extracted by a protein extraction buffer [50 mM Tris–HCl (pH 6.8), 9 M urea, 4.5% SDS and 7.5% β-mercaptoethanol]. Then, equal amounts of protein were separated on 12.5% SDS-PAGE gels and transferred onto nitrocellulose membranes. Membranes were blocked with 5% skimmed milk powder in TBS with 0.05% Tween20 for 0.5 h at room temperature, probed with specific primary antibodies against eGFP (ABclonal, Wuhan, China), FLAG tag (Sigma, St. Louis, MO, USA), or actin (ABclonal, Wuhan, China), followed by the corresponding secondary antibodies conjugated to horseradish peroxidase [Goat anti-Rabbit IgG (H + L) secondary antibody, HRP (Thermo Fisher, Waltham, MA, USA); Goat anti-Mouse IgG (H + L) secondary antibody, HRP (Thermo Fisher, Waltham, MA, USA)]. The blotted signals were visualized using chemiluminescence according to the manufacturer’s manual (GE Healthcare, Chicago, IL, USA), and signal band quantification was performed with ImageJ (https://imagej.nih.gov/ij/, accessed on 1 May 2021).

### 2.9. Bimolecular Fluorescence Complementation (BIFC) and Subcellular Localization Assays

In BiFC and subcellular localization assays, all the plasmids were transformed into the *A. tumefaciens* strain EHA105. The *N. benthamiana* leaf epidermal cells were assayed for fluorescence using a Zeiss LSM 880 or 980 Airyscan^TM^ upright laser scanning confocal microscope (Carl Zeiss Meditec AG, Jena, Germany) at 48 hpi. Subcellular localization assays were performed in 5–7-leaf-old RFP-histone 2B (RFP-H2B) transgenic *N. benthamiana* leaves [25]. The GFP fluorophore was excited with 488 nm laser lines and the emission was detected at 490–540 nm. The RFP fluorophore was excited with 561 nm laser lines and emission was detected at 588–648 nm. For subcellular assay in protoplast, the *N. benthamiana* leaves that have expressed proteins of interest by agroinfiltration at 48 hpi were cut into small pieces (2 mm × 2 mm) followed by incubation at 28 °C in enzyme solution [1% cellulose (Yakult, Tokyo, Japan), 0.5% BSA (Sigma, St. Louis, MO, USA), 1% Driselase (Sigma, St. Louis, MO, USA) and 0.55 M mannitol (pH 5.9)] for 2 h without shaking. The released protoplasts were collected and examined by confocal microscopy. The Z-stacks of optical sections were constructed to view the localization in the protoplast using ZEN Black software (Carl Zeiss Meditec AG, Jena, Germany). In the BiFC assay, NbREM1.1 or NbREM1.5 was fused into a pCAMBIA1300-35S::MCS::nYFP vector carrying the N-terminal 159 amino acids of the YFP by T4 DNA ligase (ThermoFisher, Waltham, MA, USA) generating NbREM1.1-nYFP or NbREM1.5-nYFP. NSvc4 or TMV-MP was fused into a pCAMBIA1300-35S::MCS::cYFP vector carrying the C-terminal 80 amino acids of YFP by T4 DNA ligase (ThermoFisher, Waltham, MA, USA), generating NSvc4-cYFP or TMV-MP-cYFP. Co-infiltration of *Agrobacterium* strains containing the BiFC constructs that were used to test the interaction was carried out at OD_600_ of 0.5:0.5. The NbREM1.1-nYFP and NSvc4-cYFP interaction was used as a positive control; the NbREM1.1-nYFP and TMV-MP-cYFP interaction was used as a negative control.

### 2.10. Tobacco Rattle Virus (TRV)-Induced Gene Silencing

A 350-bp *NbREM1.5* cDNA fragment was cloned into TRV-RNA2 to generate TRV-RNA2::*NbREM1.5*, used for silencing *NbREM1.5* in *N. benthamiana*. A 350-bp *mCherry* cDNA fragment was cloned into TRV-RNA2 to generate TRV-RNA2::*mCherry*, used as a negative control. TRV-RNA1, TRV-RNA2::*NbREM1.5* and TRV-RNA2::*mCherry* constructs were individually transformed into the *A. tumefaciens* strain EHA105. *A.*  *tumefaciens* cells were individually cultivated overnight at 28 °C in a YEP medium containing 50 mg/L kanamycin and 50 mg/L rifampicin. *Agrobacterium* cells harboring TRV RNA1, TRV-RNA2::*NbREM1.5,* TRV-RNA2:: *mCherry* constructs were collected by centrifugation, re-suspended to an optical density of OD_600_ = 0.2 in an infiltration buffer [10 mM MgCl_2_, 100 mM MES (pH 5.7), 2 mM acetosyringone], and used for agroinfiltration. *Agrobacterium* cells harboring TRV RNA1, equally mixed with *Agrobacterium* cells harboring TRV-RNA2::*NbREM1.5* or TRV-RNA2:: *mCherry* constructs, were syringe infiltrated into fully expanded leaves of *N. benthamiana* at the 5-week-old stage.

### 2.11. Callose Deposition Assay

Callose was stained using aniline blue fluorochrome (Biosupplies, Bundoora, Victoria, Australia) according to the user manual, with minor modification. Briefly, the *N. benthamiana* leaves that were used for aniline blue staining were cut into small pieces (4 mm × 4 mm) followed by being vacuumed for 10 min in 0.1% aniline blue fluorochrome. Leaf pieces were then observed by using LSM 880 or 980 (Carl Zeiss) and excited at 405 nm. The callose intensity was measured by using Image Pro Plus 6.0 software (Media Cybernetics, Silver Spring, MD, USA).

### 2.12. Quantification and Statistical Analysis

All experiments involving measurements, quantifications, and imaging were repeated at least three times. GraphPad Prism version 8.0 (GraphPad Software, San Diego, CA, USA) was used for data plotting and statistical tests. Statistical parameters such as mean ± SD (standard deviation), SEM (standard error), or 95% confidence intervals are indicated in figure legends. In graphs, asterisks mark statistical significance (* 0.01 < *p* < 0.05; ** *p* < 0.01; *** *p* < 0.001; ns, *p* > 0.05, not significant) according to two-tailed Student’s *t*-test.

## 3. Results

### 3.1. Group 1 Remorin Proteins Are Palmitoylated in N. benthamiana

To test whether palmitoylation exists in group 1 remorin proteins in *N. benthamiana*, we used GPS-Palm [26] to predict putative palmitoylation sites in nine members of *N. benthamiana* group 1 remorin proteins (NbREM1.1, NbREM1.2, NbREM1.3, NbREM1.4, NbREM1.5, NbREM1.6, NbREM1.7, NbREM1.8, and NbREM1.9). All of the nine group 1 remorin proteins were predicted to harbor at least one putative palmitoylation site (Appendix A). Pairwise identity analysis of nine group 1 remorin proteins showed that the identity of the amino acid sequence between NbREM1.1 and NbREM1.2, NbREM1.4 and NbREM1.5, NbREM1.6 and NbREM1.8, was 98.6%, 97.2%, and 91.7%, respectively (Appendix A). Evolutionary analysis also demonstrates that NbREM1.1 and NbREM1.2, NbREM1.4 and NbREM1.5, and NbREM1.6 and NbREM1.8 were clustered into the same clade (Appendix A), suggesting that the biological functions of NbREM1.1 and NbREM1.2, NbREM1.4 and NbREM1.5, and NbREM1.6 and NbREM1.8 proteins may be redundant. As high sequence identity was found, clade specific primer pairs were then designed to detect the transcriptional expression level of *NbREM1.1*/*NbREM1.2*, *NbREM1.4*/*NbREM1.5*, or *NbREM1.6*/*NbREM1.8* by RT-qPCR, and the specific primer pairs were designed to detect the transcriptional expression level of *NbREM1.3*, *NbREM1.7*, and *NbREM1.9* by RT-qPCR, respectively. The results showed that the native expression levels of *NbREM1.1*/*1.2*, *NbREM1.3*, and *NbREM1.4*/*1.5* were remarkably higher than that of *NbREM1.6*/*1.8*, *NbREM1.7*, and *NbREM1.9* (Appendix A). We then chose NbREM1.1, NbREM1.3, NbREM1.5, and NbREM1.8 to verify the presence of palmitoylation by acyl-resin assisted capture (acyl-RAC) [23], which is more sensitive and accurate compared to acyl-exchange chemistry [27].

NbREM1.1 has been proven to be palmitoylated [8], so we used it as a positive control for palmitoylation. In acyl-RAC assays, after the blocking of free thiols in total protein extraction with methyl methanethiosulfonate (NMT) and the cleaving of thioester linkages with hydroxylamine (NH_2_OH), palmitoylated NbREM1.1, NbREM1.3, NbREM1.5, or NbREM1.8 was captured by thiopropyl Sepharose in hydroxylamine-treated samples and detected by immunoblotting (Figure 1a–d,j). In contrast, no remarkable palmitoylation was detected in the NbREM1.1-C206A, NbREM1.3-C177A, and NbREM1.8-C195A point mutant proteins, in which the cysteine residues at the predicted palmitoylation sites were substituted by alanine residues (Figure 1e,f,i,j), indicating that the Cys206, Cys177, or Cys195 residue in the NbREM1.1, NbREM1.3, or NbREM1.8 protein is the essential target residue for palmitoylation, respectively. While in NbREM1.5, single-site mutation of Cys172 in NbREM1.5 could not completely abolish its palmitoylation (Figure 1g,j), but palmitoylation of NbREM1.5 was remarkably reduced when both Cys172 and Cys175 were simultaneously mutated (Figure 1h,j), indicating that NbREM1.5 was palmitoylated at both the Cys172 and Cys175 residues, in line with the results of the GPS-Palm prediction that NbREM1.5 has two predicted palmitoylation cysteines.

### 3.2. Palmitoylation Contributes to Remorin Protein Accumulation

To test whether palmitoylation broadly contributes to remorin protein turnover, we compared the expression level of NbREM1.1, NbREM1.3, NbREM1.5, and NbREM1.8 with their respective palmitoylation-deficient mutants. The *A. tumefaciens* expressing the flag epitope-tagged NbREMs and their palmitoylation-deficient mutants were separately infiltrated into the opposite halves of the *N. benthamiana* leaves. Immunoblotting showed that accumulation levels of NbREM1.1-C206A, NbREM1.3-C177A, and NbREM1.8-C195A were significantly lower than those of their wild-type proteins (Figure 2a–c). However, for NbREM1.5, the single-site mutant in Cys172 was insufficient to affect NbREM1.5 protein accumulation (Figure 2d), whereas its double site mutant in Cys172 and Cys175 accumulated to a remarkably lower level, compared to wild-type NbREM1.5 (Figure 2e). These results indicate that palmitoylation is broadly required for remorin protein accumulation.

### 3.3. Palmitoylation Contributes to Plasma Membrane Localization for Remorin Proteins

To test whether palmitoylation contributes to plasma membrane localization for NbREMs, wild-type GFP-NbREMs and their palmitoylation-deficient mutants were transiently expressed in *N. benthamiana* protoplast or epidermal cells. Confocal microscopy indicates wild-type GFP-NbREMs localized at the plasma membrane (Figure 3). In contrast, the plasma membrane localization of all the four NbREMs palmitoylation-deficient mutants was remarkably weakened, and their fluorescent signals were observed in nucleus and cytoplasm (Figure 3 and Appendix A). For NbREM1.5, both the single-site mutant in Cys172 or double-site mutant in Cys172 and Cys175 remarkably weakened its membrane localization, suggesting membrane localization of NbREM1.5 is more sensitive to palmitoylation deficiency (Figure 3d lower row and Appendix Ac). These results demonstrated that palmitoylation is strictly required for the plasma membrane localization.

### 3.4. NbREM1.5 Negatively Regulates TMV Cell-to-Cell Movement

To test whether *N. benthamiana* remorin proteins are involved in the regulation of TMV infection, NbREM1.1, NbREM1.3, NbREM1.5, or NbREM1.8 was transiently co-expressed with TMV-GFP, respectively. Compared with scarlet (negative control), we found that only co-expression with NbREM1.5 could significantly hamper TMV cell-to-cell movement (Figure 4a,b). To confirm whether NbREM1.5 restricts TMV cell-to-cell movement, we induced knock-down of NbREM1.5 in *N. benthamiana* plants by TRV-based gene silencing. After confirming knock-down of NbREM1.5 by RT-qPCR at 10 dpi of TRV (Appendix A), we inoculated TMV-GFP and found silencing of NbREM1.5 correlated with an increase in TMV-GFP cell-to-cell movement in inoculated leaves (Figure 4c,d), validating that NbREM1.5 negatively regulates TMV cell-to-cell movement.

### 3.5. NbREM1.5 Interacts with TMV-MP and Interferes with Its Ability for Expanding Plasmodesmata Size Exclusion Limit

BiFC assays were used to test whether NbREM1.5 could interact with the TMV-MP in vivo. Strong plasma membrane-localized yellow fluorescence was observed upon co-expression of NbREM1.5-nYFP with TMV-MP-cYFP, whereas no fluorescence was observed in the combinations of NbREM1.5-nYFP with RSV-NSvc4-cYFP, or NbREM1.1-nYFP with TMV-MP-cYFP (Figure 5a). In vivo interaction between NbREM1.1 and RSV-NSvc4 was found in our previous study [8] and was used as a positive control; the interaction between NbREM1.1-nYFP and TMV-MP-cYFP was used as a negative control (Figure 5a). These results indicate that NbREM1.5 specifically interacts with TMV-MP in vivo.

TMV-MP is known to expand the plasmodesmata size exclusion limit (SEL) [28]; we then tested whether NbREM1.5 could affect the ability of TMV-MP to expand the plasmodesmata size exclusion limit. Highly diluted *Agrobacterium* (1000×) expressing GFP was co-infiltrated with *Agrobacterium* expressing TMV-MP and the tomato bushy stunt virus encoded gene silencing suppressor P19, and then the diffusion of GFP from an individual expressing cell to adjacent cells through plasmodesmata was monitored at 3 dpi. P19 can enhance the transient expression of proteins by agroinfiltration [29]. We found that overexpression of NbREM1.5 together with TMV-MP significantly restricted the GFP diffusion to neighboring cells, as compared with the overexpression of GUS-flag with TMV-MP, suggesting that NbREM1.5 directly affects the ability of TMV-MP to expand the plasmodesmata size exclusion limit (Figure 5b,c).

The callose deposition assay by using aniline blue staining showed that callose deposition at the plasmodesmata was significantly increased in *N. benthamiana* epidermal cells after overexpression of Flag-NbREM1.5 by agroinfiltration, compared with overexpression of GUS as a control (Appendix A). These results indicate that NbREM1.5 restricts TMV cell-to-cell movement by interacting with TMV-MP in vivo and interfering with its ability to expand the plasmodesmata size exclusion limit.

### 3.6. Palmitoylation Is Required for NbREM1.5 to Inhibit TMV Cell-to-Cell Movement

To understand if palmitoylation of NbREM1.5 is required to inhibit TMV-MP expanding the plasmodesmata size exclusion limit, we co-expressed palmitoylation-deficient mutant NbREM1.5-C172/175A with TMV-MP, as compared with overexpression of GUS-flag with TMV-MP. The results showed that the rate of GFP diffusion showed no remarkable difference when TMV-MP was co-expressed with the deficient mutant NbREM1.5-C172/175A or GUS-flag (Figure 5b,c), indicating that palmitoylation is required for NbREM1.5 to restrict the ability of TMV-MP to increase the size exclusion limit of plasmodesmata.

In *N. benthamiana* epidermal cells, there was no significant difference in callose deposition at the plasmodesmata between the overexpression of the palmitoylation-deficient mutant Flag-NbREM1.5-C172/175A and the overexpression of the GUS as a negative control (Appendix A), indicating that palmitoylation is indispensable for NbREM1.5 to directly hamper plasmodesmata permeability. We then tested whether palmitoylation plays a role in the restriction of TMV cell-to-cell movement by NbREM1.5. Infection assays in *N. benthamiana* showed that the NbREM1.5 palmitoylation-deficient mutant NbREM1.5-C172/175A completely lost the ability to restrict TMV-GFP cell-to-cell movement, even promoting TMV-GFP cell-to-cell movement (Figure 5d,e). Altogether, these results provide strong evidence that palmitoylation is crucial for NbREM1.5 to inhibit TMV cell-to-cell movement.

## 4. Discussion

Palmitoylation can enhance protein membrane affinity by the addition of long-chain fatty acids, and this process is dynamic and reversible. Palmitoylation not only contributes to the membrane association of proteins but also to the regulation of protein–protein interaction and protein turnover [8,30,31]. In this study, we unravel that palmitoylation broadly exists in *N. benthamiana* group 1 remorin proteins (Figure 1 and Appendix A). By choosing four members of the *N. benthamiana* group 1 remorin proteins (NbREM1.1, NbREM1.3, NbREM1.5, NbREM1.8) as examples, we demonstrate that palmitoylation is not only crucial for NbREMs attachment to the plasma membrane, but also indispensable for keeping NbREMs protein accumulation (Figure 2 and Figure 3). We finally reveal that palmitoylation is required for NbREM1.5 to restrict TMV cell-to-cell movement (Figure 5).

Although *N. benthamiana* has nine remorin isoforms that are phylogenetically clustered in group 1, their native transcriptional expression levels are remarkably diverse (Appendix A). As a large gene family, transcriptional expression of *NbREM1.1*, *NbREM1*.2, *NbREM1.3*, *NbREM1.4*, and *NbREM1.5* are predominantly higher than *NbREM1.6*, *NbREM1*.7, *NbREM1.8*, and *NbREM1.9*, suggesting these isoforms might play major roles in plant development (Appendix A). Nevertheless, we cannot exclude that those isoforms are partially redundant or can functionally substitute for other isoforms (Appendix A).

Group 1 remorin proteins from different plant species have been reported to negatively regulate viral cell-to-cell movement; for example, StREM1.3 can inhibit cell-to-cell movement of potato virus X (PVX) and TMV, AtREM1.2 inhibits cell-to-cell movement of TRV and turnip mosaic virus (TuMV), and NbREM1.1 inhibits RSV cell-to-cell movement, but NtREM1.2 was reported to facilitate tomato mosaic virus (ToMV) cell-to-cell movement [8,9,32,33,34,35]. By screening four homologs (*NbREM1.1*, *NbREM1.3*, *NbREM1.5,* and *NbREM1.8*) of the group 1 remorin family in *N. benthamiana*, we found that only NbREM1.5 could significantly hamper TMV cell-to-cell movement by directly affecting the permeability of the TMV-MP extended plasmodesmata (Figure 5). Although NbREM1.1 has been shown to regulate PD aperture and inhibit RSV cell-to-cell movement [8], we found that NbREM1.1 could not significantly hamper TMV cell-to-cell movement (Figure 4a,b). StREM1.3 physically interacts with PVX movement protein TGB1 and impairs its ability to increase plasmodesmata permeability [28]. NbREM1.1 is also able to directly interact with RSV movement protein NSvc4 [8]. Our in vivo interaction assay by BiFC showed that NbREM1.1 cannot interact with TMV-MP, but NbREM1.5 can specifically interact with TMV-MP in the plasma membrane (Figure 5a), suggesting that the inhibition of viral motility by remorin proteins may depend on the interaction between remorin proteins and viral movement proteins.

Plasma membranes were believed to be sub-compartmentalized into diverse nanodomains [36]. Evolutionary distant groups of remorin proteins were found localizing in separate nanodomains, indicating that distinct remorin groups cluster into separate domains to play independent roles [37]. It is unclear whether different members of group 1 remorin proteins could also localize in separate nanodomains, which might lead to different roles for group 1 remorin homologs for modulating diverse virus cell-to-cell movement.

Many peripheral membrane proteins require cycles of palmitoylation and depalmitoylation for plasma membrane delivery [30]. We showed that NbREM1.5 was palmitoylated in both Cys172 and Cys175 residues (Figure 1g–j). The single-site mutant in NbREM1.5 is insufficient to affect NbREM1.5 protein accumulation (Figure 2d), whereas it is sufficient to affect NbREM1.5 plasma membrane localization (Figure 3d), indicating that, compared with protein turnover, plasma membrane localization is more sensitive to palmitoylation. Palmitoylation is required for GSD1 to regulate plasmodesmata conductance and for NbREM1.1 to restrict RSV cell-to-cell movement [8,18]. Interestingly, we showed that the palmitoylation-deficient mutant of NbREM1.5 (NbREM1.5-C172/175A) even promoted TMV cell-to-cell movement (Figure 5d,e); it is possible that NbREM1.5-C172/175A may work as a dominant-negative mutation that might interact with and interfere wild-type NbREM1.5, restricting TMV movement.

Remorin consists of a highly variable phosphorylated and intrinsically disordered N-terminal region and a conserved C-terminal region. The remorin C-terminal region contains a membrane-anchoring motif and harbors cysteine residues that can be palmitoylated [11,14,38,39]. We believe that both the C-terminal membrane-anchoring motif and palmitoylation at the C-terminal cysteine residues co-contribute to the plasma membrane association and membrane nanodomains anchoring of remorin proteins. The N-terminal region of several remorin proteins can be phosphorylated by membrane-localized protein kinase under abiotic or biotic stress [33,39,40]. StREM1.3 was reported to be phosphorylated by membrane-bound AtCDPK3, which defines its plasma membrane nanodomain organization and is crucial for StREM1.3 to restrict PVX cell-to-cell movement [33]. It is possible that the cytoplasmic-retained palmitoylation-deficient remorin proteins cannot be phosphorylated by a membrane-bound protein kinase, and they lost their ability to restrict viral cell-to-cell movement. So the interplay between phosphorylation in the remorin N-terminus and palmitoylation in remorin C-terminus deserves to be further investigated. It is also important for elucidation of the details concerning the enzymology of the remorin palmitoylation–depalmitoylation cycle process.

## Figures and Tables

**Figure 1 viruses-14-01324-f001:**
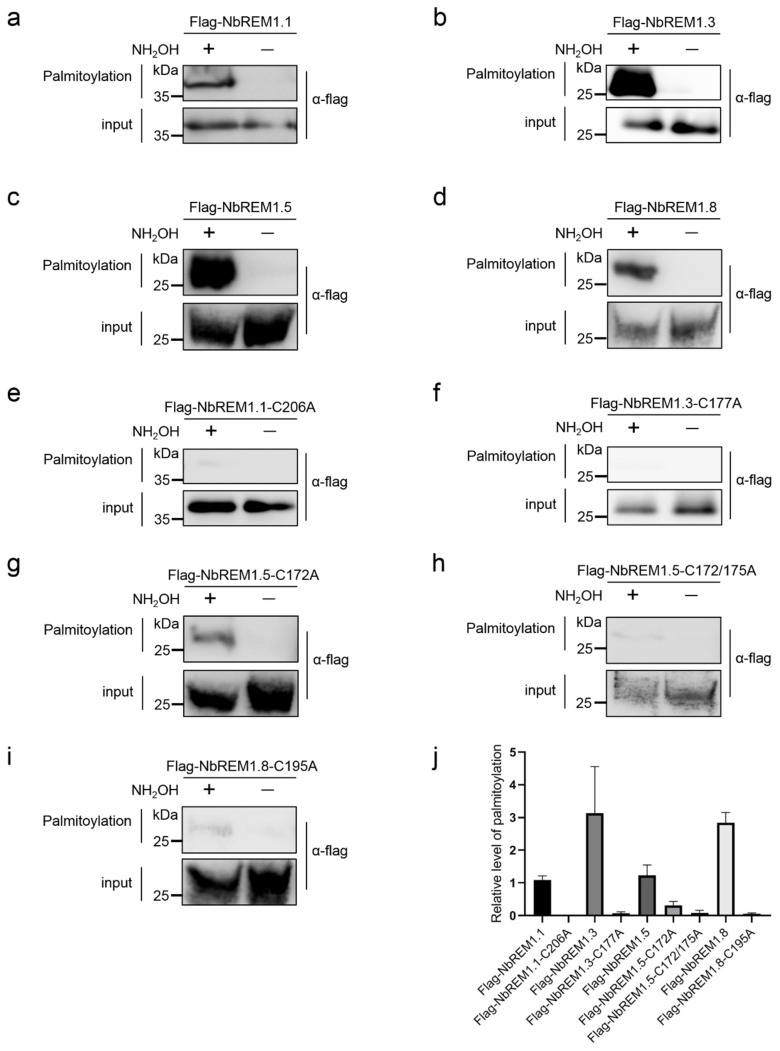
Acyl-resin assisted capture (Acyl-RAC) to detect palmitoylation of NbREMs and their respective palmitoylation-defective mutants in *N. benthamiana* leaves. (**a**) NbREM1.1, (**b**) NbREM1.3, (**c**) NbREM1.5, (**d**) NbREM1.8, (**e**) NbREM1.1-C206A, (**f**) NbREM1.3-C177A, (**g**) NbREM1.5-C172A, (**h**) NbREM1.5-C172/175A, (**i**) NbREM1.8-C195A. (**j**) The relative palmitoylation levels of NbREMs and their respective palmitoylation-defective mutants. Data are mean ± SD (*n* = 3). Immunoblotting was performed using anti-flag antibody. NH_2_OH indicates presence (+) or absence (−) of hydroxylamine required for acyl group cleavage during the thiopropyl Sepharose 6b capture step. Thiopropyl Sepharose 6b enriched proteins were eluted and represent palmitoylated proteins (palmitoylation). Prior to thiopropyl Sepharose 6b capture the samples were removed as an input loading control (input).

**Figure 2 viruses-14-01324-f002:**
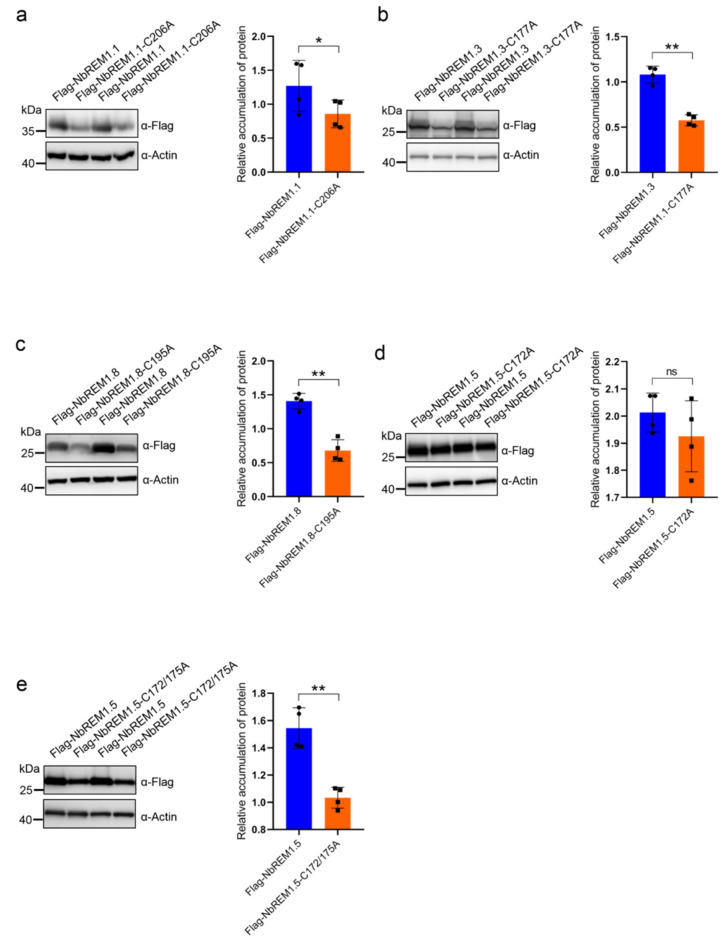
Protein accumulation assay comparison of NbREM1.1 and NbREM1.1-C206A (**a**), NbREM1.3 and NbREM1.3-C177A (**b**), NbREM1.8 and NbREM1.8-C195A (**c**), NbREM1.5 and NbREM1.5-C172A (**d**), NbREM1.5 and NbREM1.5-C172/175A (**e**) in *N. benthamiana* leaves. Data are mean ± SD (*n* = 4). Asterisks mark significant differences according to two-tailed Student’s *t*-test; * *p* < 0.05; ** *p* < 0.01; ns, no significant difference. The opposite halves of *N. benthamiana* leaves separately expressed flag-tagged NbREMs or their palmitoylation-defective mutants by agro-infiltration. Immunoblotting with anti-actin antibody was used as a loading control. Immunoblotting with anti-flag antibody was used to detect protein accumulation of NbREMs and its palmitoylation-defective mutants.

**Figure 3 viruses-14-01324-f003:**
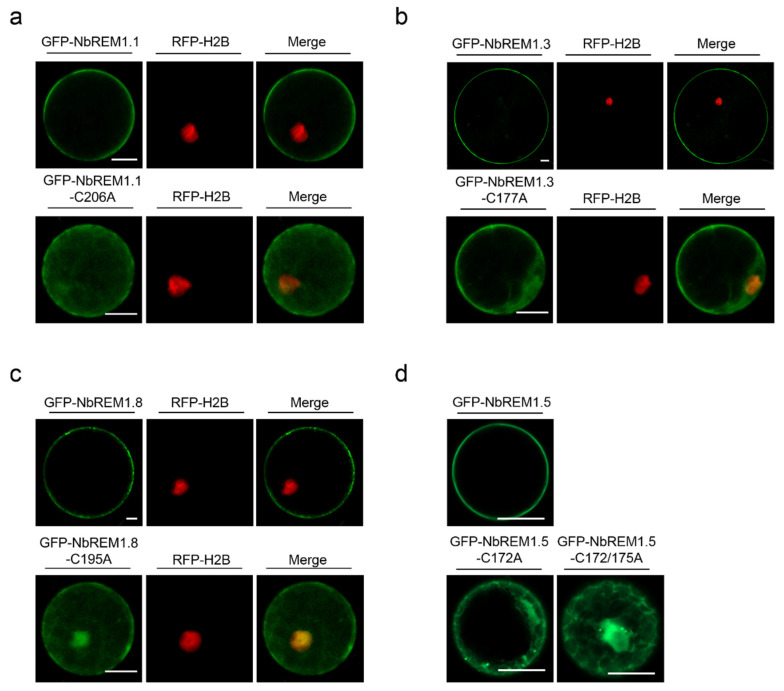
Subcellular localization of GFP-NbREMs and their respective palmitoylation-defective mutants in RFP-H2B transgenic *N. benthamiana* protoplast cells. (**a**) GFP-NbREM1.1 and GFP-NbREM1.1-C206A, (**b**) GFP-NbREM1.3 and GFP-NbREM1.3-C177A, (**c**) GFP-NbREM1.8 and GFP-NbREM1.8-C195A, (**d**) GFP-NbREM1.5, GFP-NbREM1.5-C172A, and GFP-NbREM1.5-C172/175A. The Z-stacks of optical sections were constructed to view the localization in protoplast using ZEN Black software. GFP-NbREMs and their palmitoylation-defective mutants were transiently expressed in transgenic *N. benthamiana* expressing RFP-H2B, of which the nuclei was marked by red fluorescence. The released protoplasts were collected and examined by a Zeiss LSM 880 or 980 Airyscan^TM^ upright laser scanning confocal microscope at 48 hpi. Bar, 20 μm.

**Figure 4 viruses-14-01324-f004:**
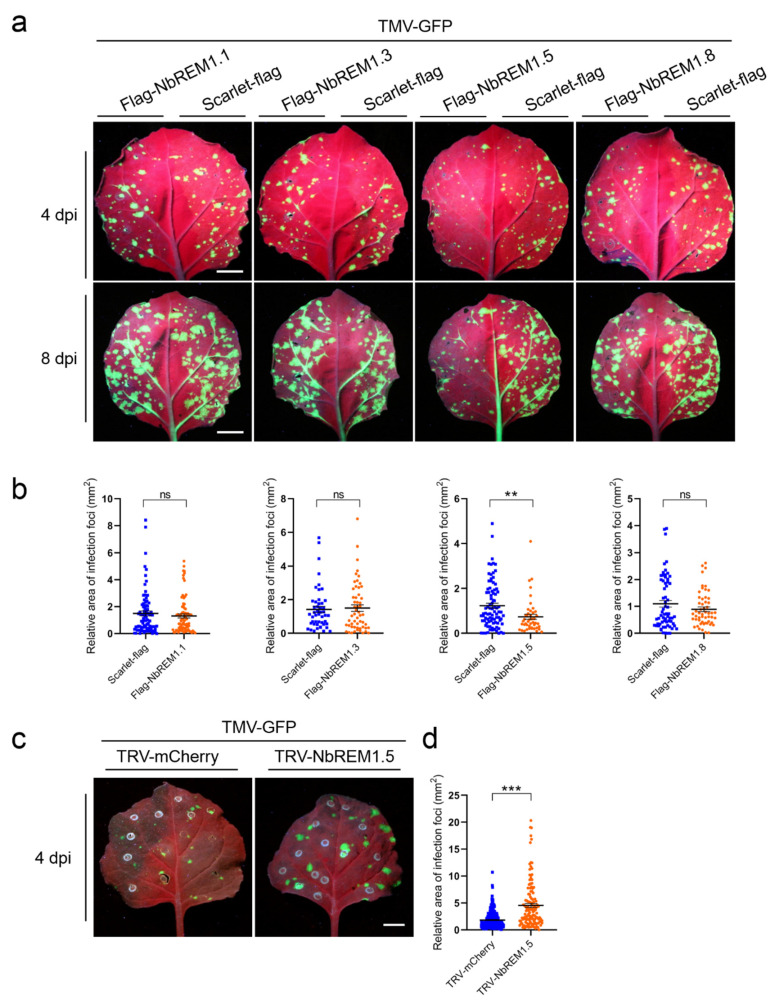
NbREM1.5 negatively regulates tobacco mosaic virus (TMV) cell-to-cell movement. (**a**) The effect of four NbREMs on TMV-GFP cell-to-cell movement at 4 days post-inoculation (dpi) and 8 dpi. TMV-GFP was co-agroinfiltrated with flag-NbREMs or scarlet-flag (a negative control) on the opposite halves of *N. benthamiana* leaves. The TMV-GFP infection foci fluorescence signals were observed at 4 dpi for upper row and 8 dpi for lower row. Leaves were captured under a portable UV lamp. Bar, 1 cm. (**b**) Statistical analyses of the effects of the four NbREMs on TMV-GFP cell-to-cell movement in the lower row of (**a**). Data are mean ± SEM (*n* = 50). Asterisks mark significant differences according to two-tailed Student’s *t*-test; ** *p* < 0.01; ns, no significant difference. (**c**) Detection of the infection foci of TMV-GFP in TRV-mCherry (control) and NbREM1.5 knock-down (TRV-NbREM1.5) *N. benthamiana* plant leaves. The TMV-GFP infection foci fluorescence signals were observed at 4 dpi under the portable UV lamp. (**d**) Statistical analyses of TMV-GFP infection foci area in TRV-mCherry (as a control) and NbREM1.5 knock-down (TRV-NbREM1.5) *N. benthamiana* plant leaves. Data are mean ± SEM (*n* = 133). Asterisks mark significant differences according to two-tailed Student’s *t*-test; *** *p* < 0.001.

**Figure 5 viruses-14-01324-f005:**
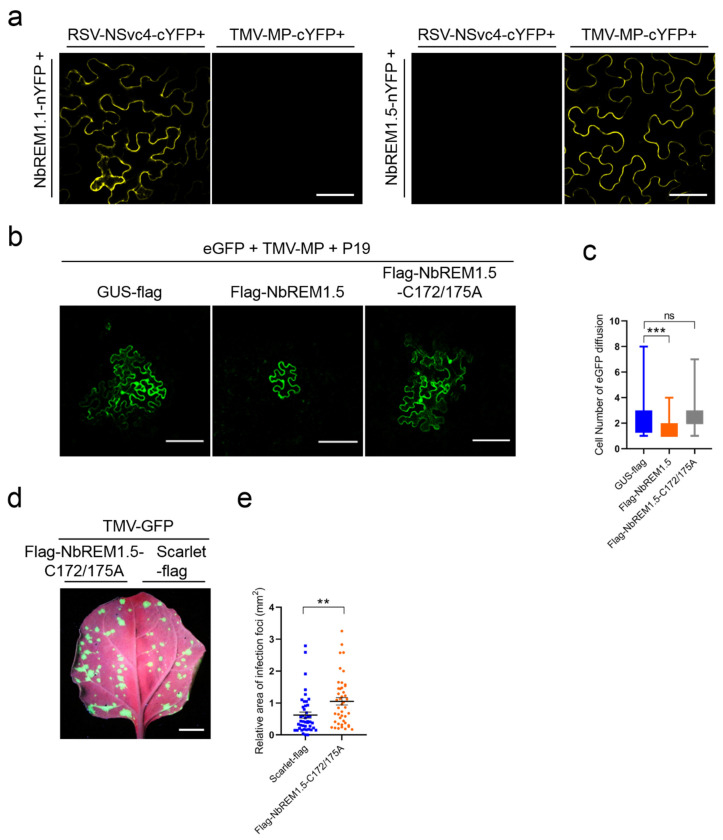
Palmitoylation is indispensable for NbREM1.5 to negatively regulate TMV cell-to-cell movement. (**a**) NbREM1.5 interacts with TMV MP in vivo by BiFC. Reconstituted YFP signals were observed at 48 h post-inoculation (hpi) by fluorescence confocal microscope in *N. benthamiana* leaf epidermal cells. Bar, 20 μm. (**b**) Effect of NbREM1.5 and its palmitoylation-defective mutants NbREM1.5-C172/175A on cell-to-cell diffusion of GFP under co-expression with TMV-MP. GUS-flag was used as the negative control. Confocal images were taken at 48 hpi. Bar, 20 μm. (**c**) Statistical analyses of the effect of NbREM1.5 and NbREM1.5-C172/175A on cell-to-cell diffusion of GFP under co-expression with TMV-MP. Data are mean ± SD (*n* = 65). Asterisks mark significant differences and “ns” marks no significant difference according to two-tailed Student’s *t*-test; *** *p* < 0.001. (**d**) Effect of NbREM1.5-C172/175A on TMV-GFP cell-to-cell movement. TMV-GFP was co-agroinfiltrated with flag-NbREM1.5-C172/175A or scarlet-flag (as a negative control) on the opposite halves of *N. benthamiana* leaves. The TMV-GFP infection foci fluorescence signals were observed at 4 days post-inoculation under a portable UV lamp. Bar, 1 cm. (**e**) Statistical analyses of the effect of NbREM1.5-C172/175A on TMV-GFP cell-to-cell movement. Data are mean ± SEM (*n* = 45). Asterisks mark significant differences according to two-tailed Student’s *t*-test; ** *p* < 0.01.

## Data Availability

All the data used in this study are already provided in the manuscript in its required section. There is no underlying data available.

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
