# Peer review of "Palmitoylation Is Indispensable for Remorin to Restrict Tobacco Mosaic Virus Cell-to-Cell Movement in Nicotiana benthamiana"

_viruses, 2022, doi:10.3390/v14061324_

Round 1

Reviewer 1 Report

This manuscript is well revised and almost ready for publication. However, there are some corrections that need to be made  and one more suggestion that are listed below. I would like the authors to revise the manuscript fully.

1. Line 48: inter-changeable may be inter-changeably.

2. Line 81: Insert "and" between NbREM1.5 and NbREN1.8.

3. Line 124: Insert "and" between NbREM1.4/NbREM1.5 and NbREM1.6/NbREM1.8.

4. Lines 189, 230, and 332: I think that the confocal microscope is either 880 or 980 and may not be 880/980. Please check the  model actually used by the authors.

5. Line 398: Correct contro to control.

6. The revised manuscript clearly explains how to make and obtain the plasmids used in the study, as I suggested. However, only the information on GUS-flag is missing. I would like the authors to explain how the plasmid was obtained.

Author Response

  1. Line 48: inter-changeable may be inter-changeably.
  2. Line 81: Insert "and" between NbREM1.5 and NbREN1.8.
  3. Line 124: Insert "and" between NbREM1.4/NbREM1.5 and NbREM1.6/NbREM1.8.
  4. Lines 189, 230, and 332: I think that the confocal microscope is either 880 or 980 and may not be 880/980. Please check the model actually used by the authors.
  5. Line 398: Correct contro to control.

Our response: Al the above parts were revised accordingly.

  1. The revised manuscript clearly explains how to make and obtain the plasmids used in the study, as I suggested. However, only the information on GUS-flag is missing. I would like the authors to explain how the plasmid was obtained.

Our responses: The information on GUS-flag is included now.

Reviewer 2 Report

Dear Authors,

I have an opportunity to review manuscript entitled: “Palmitoylation is Indispensable for remorin to restrict Tobacco Mosaic Virus cell-to-cell movement in Nicotiana benthamiana” resubmitted to the Viruses in MDPI.

Authors concentrated on remorin (REM) as a plant-specific plasma membrane-associated protein, which are bale to regulate plasmodesmata plasticity and also has an influence on restriction viral cell-to-cell movement.

Althought the role of protein S-palmitoylation in plant immunity is known, and the role of remorin was previously presented mainly in Potexvirus biology, article is interesting and provides new knowledge to the understanding of plant-virus interations. First of all, Authors screening the four members of N. benthamiana group 1 remorin proteins and found that only NbREM1.5 could significantly hamper tobacco mosaic virus (TMV) cell-to-cell movement. Further studies showed, that NbREM1.5 interacts with movement protein of TMV in vivo and induces expanding plasmodesmata size exclusion limit.

·       In my opinion the introduction the provided sufficient background for the reader;

·       Materials and methods are significantly improved,

·       Results are well described, but I suggest to fit figure 2 to the whole page, because in current form the reader lost important details;

·       Supplementary materials constituting important attachments of data- unfortunately, in figure S3 is an error – leaf, stem, flower are not plant tissues (!!) it is plant organs;

·       In figure S4 should be” localization in epidermis” instead of “Figure S4. Subcellular localization of NbREMs”-because Authors check it only in cells of one plant tissue;

·       I found some mistakes in references list, please check it carefully;

I suggest also add future prospects coming from obtained results as a kind of conclusion underlined deeply the important meaning of presented studies to the wider audience.

Author Response

  • Results are well described, but I suggest to fit figure 2 to the whole page, because in current form the reader lost important details;

Our response: We will ask the press to make the figure 2 to fit the whole page.

  • Supplementary materials constituting important attachments of data- unfortunately, in figure S3 is an error – leaf, stem, flower are not plant tissues (!!) it is plant organs;

Our response: Revised accordingly.

  • In figure S4 should be” localization in epidermis” instead of “Figure S4. Subcellular localization of NbREMs”-because Authors check it only in cells of one plant tissue;

Our response: Revised accordingly.

  • I found some mistakes in references list, please check it carefully;

Our response: I checked the references carefully.

I suggest also add future prospects coming from obtained results as a kind of conclusion underlined deeply the important meaning of presented studies to the wider audience.

Our response: We add a sentence ‘It is also important for elucidation of the details concerning the enzymology of the remorin palmitoylation-depalmitoylation cycle process’.

This manuscript is a resubmission of an earlier submission. The following is a list of the peer review reports and author responses from that submission.

Round 1

Reviewer 1 Report

In this study, Ma et al. performed extensive functional analyses of group 1 remorin proteins to examine the effects of palmitoylation on protein stability, plasma membrane localization, and inhibitory effect on TMV cell-to-cell movement, and MP-mediated SEL expansion. Their experiments are very well designed and the results, including supplementary data, are very clearly shown. I think this paper is well worth publishing in this journal. However, I have questions about the presentation and interpretation of Figure 1, as well as comments on unclear points on experimental procedures and materials and also some minor errors. The authors need to address and revise them before publication.

  1. My biggest concern is the authors' conclusion that mutants of NbREMs are not palmitoylated. To me, it appears that faint signals were detected in the presence of NH2OH. The f image in particular has a stronger white balance than the others, but a faint band is visible when I adjust the contrast of the image on my PC. I do not think the authors need to claim strongly that these mutants are not palmitoylated, but it is enough to mention that the examined cysteine is the essential target residue for palmitoylation. I hope that the authors will present the image as it is and describe only the facts in a straightforward manner.
  2. Abstract mentions that NbREM1.5 interacts with TMV MP in vivo, but that data is included in the supplemental data, not in the text. If the protein-protein interaction result is to be included in the abstract, the related images shown in Figure S7 should be moved to the text. Otherwise, the related sentence in the abstract should be rewritten.
  3. Line 68: The procedures of plasmid construction should be described in more detail in the text or provided as supplementary material. Without the specific information of vectors and primers, no one will be able to reproduce their plasmids. Such information would also be useful for the reader to understand more clearly whether the epitope tag or fluorescent protein is fused to the N- or C-terminus.
  4. Lines 78, 157 and 158: The compositions of the infiltration buffers are different. Please check the correct composition.
  5. Line 87: Please describe the information of NbAction used as a reference gene for RT-qPCR, including its accession number or reference.
  6. Line 132: Please add the plasmid information of H2B-RFP, including the source of the plasmid or its reference.
  7. Line 303: Please explain why p19 was used in the experiment. What would have happened in the absence of p19?
  8. Line 306: NbREM1.5 should be TMV-MP.
  9. In Figure 3, the images of mutants seem projective views while those of wt NbREMs single views. Please describe how these images were taken.
  10. The plasmid information of scarlet-flag shown in Figure 4 should be added to Materials and Methods.
  11. There are expressions that I would like the authors to check and reconsider.
  • Line 71: proteind may be proteins.
  • Line 95: infiltrate would be infiltration.
  • Line 105: thiods may be thiols.
  • Line 197: remarkable would be remarkably.
  • Line 200: accuracy would be accurate.
  • Line 267: transiently would be transient.
  • Line 268: co-expressed would be co-expression.
  • Line 275: negative would be negatively.
  • Line 410: “due to” is followed by a noun phrase, so please rephrase the sentence.
  • Line 422: qRT-PCR would be RT-qPCR for consistency in the text.

Reviewer 2 Report

This study investigates the role of remorins in virus movement, specifically the contribution of palmitoylation to protein accumulation and its function.  Remorins are known to be important for virus movement, but there are few studies to characterize these genes.  Working in N. benthamiana can be challenging because while these genes are mapped, it is not clear if they are cloned prior to this study.   This study appears to have cloned them, but does not report accessions indicating the gene sequences were deposited in a public repository.  I am curious to how similar the cloned genes are to the database reported sequences or if they noticed differences.  

M&M needs to be more elaborate.  Sect 2.1 indicates that transgenic plants were used, but does not list them.  Are the homozygous, what T generation?   Expression levels?  were the reported before or are they characterized here for the first time?   The backbone for cloning remorins is not provided and the binary vector used is not reported.  AN empty binary vector or heterologous gene is an important control for all studies and does not appear to be used in any of the figures. Line 80 says the primers used for cloning are listed in table S1, but they are not listed.  

Section 2.2 identifies a tree but should indicate the figure. 

Section 2.3   RNA was extracted from whole plants or leaves?   How many days post infiltration?   RT-qPCR need to be better explains as to the targets and control targets used.  Was ddCT method or another method used.  What statistics?  What software?  

Section 2.4--which TMV-GFP infecitous clone--need reference?  Is it TRBO or another?  IF this is an original plasmid then explain its construction and the strain/isolate of TMV used.  Was statistics sued for GFP analysis?  

section 2.5.   Line 100 is poor grammar.  Second sentence indicates no controls were harvested or used in the Acyl-RAC study.  The grammar in this section needs improving.  Positive and negative controls are essential.  

Section 2.6  what is NC membranes?  

Section 2.7  What controls, what plasmid backbones, etc.   If these were the result of cloning the cloning procedures need to be listed above.  Names of plasmids should be provided.  BIFC controls need to be listed.  

Results.   Please check all verb tenses.  Note clustered together is redundant-clustered is fine.  Also I would argue that the functions of REM1-8 are redundant--a better synonym could be similar or overlapping.  Their promoters could control tissue specific, developmental, or hormone responses.  Their co=factors could influence functions.  

3.1  Spell out numbers one through ten.  Where ever there is 9 it should be spelled out.  There needs to be a reference for GPS-Palm. Supplementary Figure 1 is a table and should be moved out of supplement into forefront as Table 1 or 2.  Line 191, what does redundant primers mean?   their names and reference to table needs to be explicit.  line 194-are these gene specific or group specific primers?  

The acyl_RAC assays are central to this work and these lack critical details:   experimental details are lacking.   Leaves were infiltrated with agro?   how many days later were proteins extracted?   Did you perform controls such as agro with plasmid backbone, agro with GFP (Positive control for gene expression) or infiltration buffer alone.  These additional controls will show the specificity of the acyl-RAC assays.  Figure 1 needs positive control for palmitoylation.   Blots should show total sample as well as palmitoylated and the % modification should be calculated versus control. 

The Cys modified samples showed reduced palmitoylation-potentially, however we do not know the % change in modification relative to the total. There are clear bands in gels to show that these Cys are not the only residues that are modified, or the assays are contaminated.  Perhaps you would get a similar result from healthy tissue?  

Sec 3.2 talks about protein stability, but the assay measures accumulation.  One could argue the change in accumulation is due to turnover but that is inferred.  Need to perform densitometry and statistics to make this assertion.  Clearly modification is not eliminated by Cys mutation.   there are no controls for figure 

Figure 3 needs marker for plasma membrane.  

Figure 4 indicates an 8 dpi but M&M indicates an earlier time point.  Why not do multiple time points to capture cell-to-cell movement?  at 8 days virus is already systemic clearly.  

Reviewer 3 Report

Ma et al describe the characterization of four remorin proteins. First, they predicted putative palmitoylation sites in 9 members of N. benthamiana group 1 remorin proteins. They then measured the levels of the transcripts of these proteins by qRT-PCR. The highest expressed transcripts and mutant derivatives were then expressed by agroinfiltration in N.benthamiana leaves and their levels of palmitoylation assessed by acyl-RAC assays. NbREM1.5 was found to be palmitoylated at both Cys172 and Cys175 residues. Selected REMs were then screened for their stability in agroinoculation assays and cellular distribution in protoplasts. Results suggest that palmitoylation-defective mutants are distributed across the cell while the wild-types localize to the cell membrane. The effect of knocking out REM1.5 on virus movement was tested on TMV using the VIGS TRV vector and results suggested REM1.5 was the only REM that showed a significant effect on TMV infection foci areas if up or down regulated was REM1.5.

The research makes sense and aside from a few omissions (see below) is well explained and merits publication in Viruses.

The English needs a thorough check throughout as there are multiple errors of missing articles and incorrect word usage. To illustrate the problem here are some errors picked up on the first page.

13 – “is broadly presented” not sure this is the intended meaning –“is broadly present”?

17 - the movement protein of TMV in vivo and interferes with its function of expanding

18 – “We finally demonstrate” not sure this is the right wording – “We also demonstrate…”

24 – “nanopores connecting adjacent plant cells”

29 – “plant viruses encode movement proteins (MP) to expand PD permeability and promote

31 – “Several plant proteins have also been found capable of controlling…”

Other issues…

148 – change in font size?

303 – single mention of “gene silencing suppressor P19” – explain more. Why is it being used here?

The abstract needs more information on the details of the study.

Figure 2 – I’m guessing as stated in the methods section that these experiments were repeated three times? Please provide some statistics (and maybe a normalized histogram) to support the statements in 3.2. The statement 235-6 “These results indicate that palmitoylation is broadly required for remorin proteins to keep (maintain) stability” is not demonstrated by these results.

Fig 3 – Did the authors measure the levels of REMs in the protoplasts? Visually there appears to be more signal in the mutants which contrasts with the results in agroinfiltrated leaves in Fig 2. Again, the statement: Together, these results demonstrated that palmitoylation is broadly present in N. benthamiana remorin proteins, and palmitoylation is strictly required for the plasma membrane localization.” is not supported by these specific results, rather “These results suggest that palmitoylation is required for the plasma membrane localization.”

Fig 5a – supports the narrative but there should be some statistics here as to number of replicates and cells infected with each Flag-construct.